# Prevalence and correlates of overweight and obesity among primary school children in Kilimanjaro, Tanzania

**Mary Vincent Mosha**[1] *, **Sia E. Msuya**[1], **Elizabeth Kasagama**[1], **Philip Ayieko**[2¤], **Jim Todd**[1,2], **Suzanne Filteau**[2]

**1** Institute of Public Health, Kilimanjaro Christian Medical University College (KCMUCo), Moshi, Tanzania,
**2** London School of Hygiene and Tropical Medicine (LSHTM), London, England

¤ Current address: Mwanza Intervention Trials Unit (MITU), Mwanza, Tanzania
* maryanfort@yahoo.com

**Data Availability Statement:** All relevant data are within the paper and its Supporting information files.

## Abstract

### Background

Prevalence of childhood overweight and obesity in low- and middle-income countries is on the rise. We focused on multiple factors which could influence body mass index.

### Methods

A cross sectional school-based study was conducted in Moshi, Tanzania. Primary school children aged 9–11 years were recruited from 20 schools through a multistage sampling technique. Questionnaires were used to collect information on physical activity and diet by food frequency questionnaire. Height and weight measurements were taken and body mass index z scores for age and sex (BMIZ) calculated using the WHO AnthroPlus. Children were considered thin if BMIZ was <-2 standard deviations and overweight or obese if BMIZ was >1 SD. Information on school policies and environment was obtained from headteachers. Correlates of overweight and obesity were examined using a multinomial multilevel logistic regression.

### Results

A total of 1170 primary school children, of whom 636 (54%) were girls, were recruited from 20 schools. The prevalence of overweight and obesity was 15% overall (overweight 9% and obesity 6%) and most prevalent in urban areas (23%) and in private schools (24%). Moreover, thinness was found to be (10%) overall, most prevalent in rural areas (13%) and in government schools (14%). At school level, residing in urban (adjusted relative risk ratio [aRRR] 3.76; 95% confidence interval [CI] 2.49,5.68) and being in private school (aRRR 4.08; 95% CI 2.66,6.25) were associated with a higher risk of overweight and obesity while availability of playgrounds in schools (aRRR 0.68; 95% CI 0.47, 0.97) was associated with a lower risk of overweight and obesity. At home level, availability of sugary drinks (aRRR 1.52; 95% CI 1.01,2.28) was associated with a higher risk of overweight and obesity.

**Funding:** Mary Vincent Mosha (MVM) is supported by DELTAS Africa Initiative grant # DEL-15-011 to THRiVE-2. The DELTAS Africa Initiative is an independent funding scheme of the African Academy of Sciences (AAS)'s Alliance for Accelerating Excellence in Science in Africa (AESA) and supported by the New Partnership for Africa's Development Planning and Coordinating Agency (NEPAD Agency) with funding from the Wellcome Trust grant # 107742/Z/15/Z and the UK government. The views expressed in this publication are those of the author(s) and not necessarily those of AAS, NEPAD Agency, Wellcome Trust or the UK government.

**Competing interests:** The authors have declared no competing interests exist.

## Conclusion

Overweight and obesity are common in private schools and in urban settings. Efforts should be taken to ensure availability of playgrounds in schools and encouraging children to engage in physical activities.

## Introduction

The prevalence of overweight and obesity is increasing globally and across different age groups, from young age to adulthood. The World Health Organization (WHO) estimates that 43 million children are either overweight or obese [1]. In Africa, the number of children who are overweight or obese has doubled from 5.4 million in 1990 to 10.4 million in 2014 [2]. WHO set the global targets that there should be no more increase in childhood obesity by 2025, and thus countries should take actions towards prevention of childhood obesity [3].

Obesity has become a major nutrition related disease with negative impacts on health. Most children who are overweight or obese experience some health problems such as breathing difficulties, increased risks of accidents e.g., fractures, and are susceptible to many non-communicable diseases such as diabetes. Childhood obesity may persist into adulthood and increases the risks of many co-morbidities and of premature mortality [4].

A change in lifestyle which is accelerated by economic growth and urbanization influences a change in behavioural and environmental factors that expose children to an "obesogenic environment", that is a complex imbalance between calories consumed versus calories expended [5,6]. Children may have poor diets which are energy-dense and limited opportunities for regular physical activity.

The causes of obesity are complex and require a thorough understanding of its determinants at different levels. In 2005, the Institute of Medicine (IOM) *"Health in the Balance"* introduced a framework for action towards understanding childhood obesity using a socio-ecological perspective [7]. The framework postulates that child health is not affected only by personal factors (age, sex, genetics, weight status) but there are interactions from other multiple factors such as home, neighbourhood environment, schools and community "behavioural settings" which might influence healthy behaviours and affect the energy balance. For example, the built environment such as walkways plays a role in promoting safe walking and opportunities for physical activity while going to and from schools. Availability of certain types of food at home, school and the neighbouring community may promote good or bad eating behaviours [8]. In most cases, children's food choices and opportunities to engage in physical activity are determined by parents at home, school and the neighbourhood environment.

In Sub-Saharan Africa, studies on childhood overweight/obesity reported the prevalence ranges from 13%– 21% [9–15]. There is a need for more research to understand the underlying correlates. Many Sub-Saharan African countries are experiencing a shift in dietary behaviours and low physical activity levels [16,17].

Previous studies in our setting reported on prevalence of overweight and obesity without focusing the multiple factors to understand all other important correlates, and most of these studies focused on urban children. Tanzania, like many other low- and middle-income countries is experiencing a transition in which overweight and obesity occurs parallel with under nutrition. It remains unclear which correlates are important for understanding childhood overweight and obesity in both rural and urban setting.

Therefore, the current study uses the socio-ecological model [7,18] as a guide to examine the correlates of overweight and obesity for school children, aged 9–11 years, from multiple levels of influence, in order to inform the development of intervention approaches that can be useful for primary school children in Tanzania.

## Methods

### Study setting

A school-based cross-sectional study was conducted from August to November 2019 in Kilimanjaro region, northern Tanzania. Kilimanjaro region is one of the 31 administrative regions in Tanzania and is composed of seven administrative districts. Two districts of Kilimanjaro region were purposely selected, Moshi municipal and Moshi rural to represent urban and rural settings. Moshi municipality has a total of 48 primary schools of which 35 are public and 13 are private with a total number of 33,207 students while Moshi rural has a total of 272 schools of which 252 are government and 20 are private with a total number of 81,297 students. The primary school enrolment starts from age 7 for standard one to 13 years for standard seven.

### Study population

The study involved primary school children aged 9–11 years (corresponding to classes 4, 5 and 6) who were recruited from private and government schools in Moshi municipal and Moshi rural districts. We selected this age band because by the age of 9, children are generally able to read, write and express themselves and this reduced the complexities in completing the questionnaires. Also, we defined age 11 as the upper limit for our sample because there is a lot of change over the ages 6 to 15 years. The age range 9–11 represents a group of children who are more homogenous in their physical activity. We excluded children from standard 7 as they were in preparation of their final year exams. Children from boarding schools were excluded as they tend to follow the same activity and diet routine and we could not examine other lifestyle factors outside the school. Children with disabilities e.g., two children with knee joint contractures were excluded as this could interfere with anthropometry. Also, 24 children were excluded as they could not read, write, or understand the questions and express themselves.

### Sample size and sampling procedure

The sample size was estimated by considering the prevalence of overweight and obesity (14.7%) from previously collected data in Tanzania [11]. We further assumed the margin error/precision of +/-2.5% and considering a design effect of 1.3 to allow for the fact that participants are recruited from clusters (schools), and this resulted in the total estimated sample size of 1008 school children. We further added a 20% of the sample size to account for non-response, giving the final sample size of 1170, recruited from 20 schools (10 urban and 10 rural).

We employed a multistage sampling to select the study participants, randomly selecting eight wards from a list of all wards in the two selected districts. Primary schools from the selected wards were stratified by type of schools (private and government). We considered being in government or private school as a proxy indicator for lower or higher socio-economic status. Simple random sampling was used to select schools to be involved in the study. A list of all students aged 9 to 11 years corresponding to class 4–6 was generated from the attendance registers, followed by a random selection of children within each school. We selected children from schools according to probability proportional to size of schools.

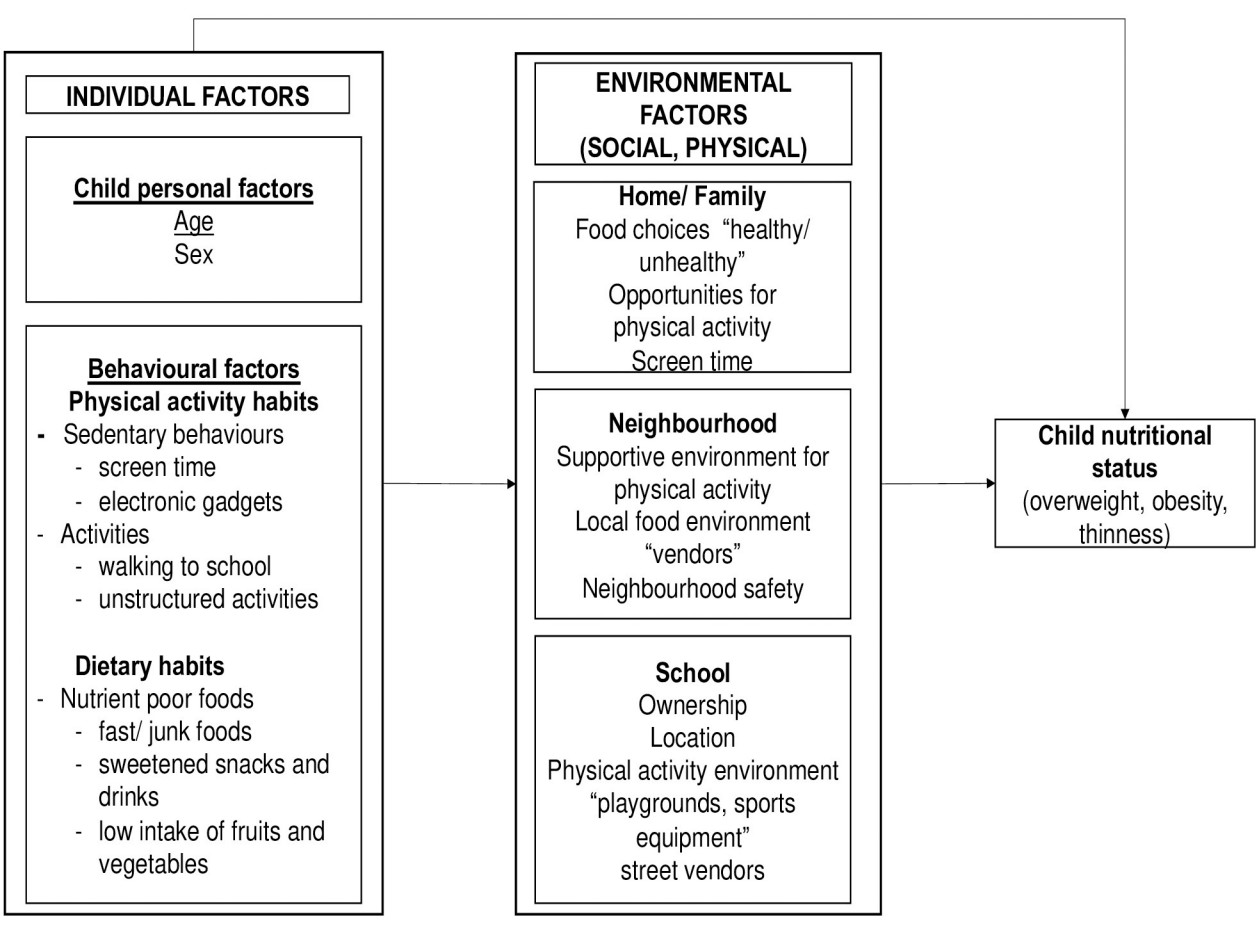

**Fig 1. Socio-ecological model for childhood overweight and obesity.**

## Study variables

Our main study outcome variables were prevalence of overweight and obesity that was obtained from the BMIZ for age and sex, and a variety of factors (correlates) possibly associated with overweight and obesity. This study used a simplified theoretical framework adapted from previous studies and modified [7,18] as a guide to examine the correlates of overweight and obesity. We focused on the proximal factors (individual/child level factors: child characteristics and behavioural factors) to distal factors (social and physical level factors), Fig 1.

## Data collection methods and tools

**Child questionnaire administration process.** This study adapted and modified the questionnaire from the International study of childhood obesity, lifestyle and environment (ISCOLE) [19] which aimed at determining the relationships between overweight/obesity and lifestyle behaviours. The modification involved removing some questions which were not applicable to the Tanzanian cultural context, thereafter the questionnaire was translated into local language (Swahili) and back translated into English to ensure fidelity to the original meaning. The modified questionnaire was designed to collect information on all child/individual, school, home and neighbourhood level factors, and it was piloted among 50 primary

school children. S1 and S2 presents English and Swahili questionnaires. Upon arrival in schools the research team was introduced to children to build a rapport and ask for children written assent. Thereafter, instructions on how to fill the questionnaires were read to children. The Research team guided children step by step to ensure that they understood each question well and answered them independently.

*Assessment of factors for overweight and obesity*. We considered multiple contexts for childhood overweight/obesity that are relevant to the socio—ecological model. Therefore, this study examined individual level characteristics, school, home and neighbourhood characteristics which may predispose children to change their behaviours and have shown to affect their nutritional status [7,20].

**Individual level factors.** *Child characteristics and behavioural factors*. Questions were asked about demographics, physical activity, e.g., participation and time spent in regular activities such as walking to school, exercise during school breaks and household chores. Sedentary behaviour related questions involved reading or sitting idle and watching television. Walking to school related responses were categorized based on the normal walking time to school i.e., "≤ 30 minutes" and "> 30 minutes". A review on sedentary behaviours, recommended less than 2 hours a day of sedentary time e.g., watching television for less than 2 hours a day. This observation guided our decision to dichotomize the responses for sedentary behaviour as "≤ 2 hours" and "> 2 hours" [21].

Diet was estimated using a 15-item food frequency questionnaire asking about frequency of intake of certain food items with the following responses: *"never", "once per day", "more than once per day", "3 to 6 times per week", "once or twice per week", "everyday more than once"*. From these responses we estimated servings per day as follows: never = 0 servings/day, once per day = 1 serving per day, 2–3 times a day/everyday more than once = 2.5 servings per day, 3–6 times per week = 0.64 servings per day, once or twice per week = 0.21. The calculated servings per day were then classified into terciles (low, medium and high) [22].

**Home level factors.** Questions included the availability of healthy versus unhealthy foods at home e.g., vegetables and fruits, sugary/sweetened drinks, fast foods and opportunities for regular physical activity. Information on the availability of media such as electronic gadgets e.g., video games and tablets, access to television and existence of mobile phones owned by any family member was collected.

**Home neighbourhood level factors.** Questions were about the home neighbourhood environment such as availability of street vendors, availability of recreational places or playgrounds for promoting and encouraging physical activity, and safe environment for children to be free to play or walk outside (environmental safety).

**School level factors.** A school environment questionnaire was administered to headteachers/administrators to collect general information on school policies, availability of sports equipment and playgrounds in schools and physical education classes.

**Anthropometric measures.** Anthropometry was done in duplicate following NHANES standard procedures [23]. Two trained research assistants took independent anthropometry readings. Routinely, we assessed the technical error of measurement (TEM) [24] to estimate the inter-observer anthropometric measurements errors.

Children were asked to remove shoes, any items from pockets, jewellery and excess clothing other than their regular school uniform. Weight was measured using daily calibrated weighing scale (TANITA model DC 430 MA), the machine was set to deduct 0.5kg (clothing weight) before starting measurements. Height was measured to the nearest 0.1 cm using the TANITA height rod with children standing straight with the head in the Frankfurt plane.

## Data processing and analysis

We performed data cleaning and analysis using STATA version 15.1 (StataCorp, College Station, TX, USA). Descriptive statistics of study participants were summarized using means and confidence intervals for continuous variables, and frequency and percentages for categorical variables.

We used the WHO AnthoPlus package (version 1.0.4) macros file for STATA SE to determine body mass index z-scores (BMIZ) according to age and sex. Children were categorized using the WHO cut off points; thin <-2 SD, normal between -2SD and 1SD, overweight >+1SD and <+2SD, obese ≥ +2SD [25].

We used a multinomial multilevel logistic regression model accounting for cluster (schools) to estimate the relative risk ratio (RRR) and 95%CI for the correlates of overweight and obesity (combined), and for thinness, considering children with normal weight as a reference group. We performed variable selection manually starting with the model containing all variables significantly associated with overweight and obesity (p<0.05) in the crude/unadjusted analysis. The next step entailed further removing variables with no statistical evidence of association with overweight and obesity until the final model was obtained. Age and sex were considered as a priori confounders, hence retained in an adjusted model at each step.

## Ethical approval

Ethical approval was granted by the National Institute for Medical Research, certificate number: IX/2735 and the Kilimanjaro Christian Medical University College Ethics Committee, certificate number: 2225. Prior to school visits, permission to interact with schools was obtained from the Regional Medical Officer, district education officers and school authorities. We thereafter visited selected schools to introduce the study to headteachers/administrators and provide them with the study information sheet and consent form. Headteachers/administrators sent information leaflets home with children explaining the study and the day of the visit. We experienced difficulties in obtaining a written parental consent as some parents did not return the consent forms that we had sent to them through their children, but instead only gave verbal consent. To address this problem, we consulted the ethics committee which provided a waiver because the study did not involve invasive procedures. Therefore, the committee allowed headteachers/administrators to consent for school children. Upon the research team's arrival at schools, headteachers/administrators introduced them to children to build a rapport. Study aims were shared with children and children were invited for any questions or concerns. Children had the right to or not to participate in the study. Written assent was obtained from children.

## Results

A total of 1170 children aged 9–11 years from 20 primary schools were enrolled in the study. Mean age of study participants was 10 (SD = 0.8) years and 636 (54%) were girls. Overall, the prevalence of overweight and obesity was 15% (overweight 9% and obesity 6%) and of thinness was 10%.

Table 1 illustrates the demographic and unadjusted behavioural correlates (physical activity, sedentary and dietary behaviours) of the study participants. Overweight and obesity prevalence was similar in boys and girls. Children who exercised during school break time were at lower risk of overweight or obesity than those who did not exercise (RRR 0.61, 95%CI 0.41, 0.89). A relatively long duration of walking to school was associated with thinness: 15% of children who walked to school for > 30 minutes were thin compared to 10% of children who walked for ≤ 30 minutes (RRR 1.86, 95% CI 1.10, 3.13). Children who reported a medium level

**Table 1. Child demographic, physical activity and dietary behavior correlates for overweight and obesity (N = 1170).**

| Characteristics | N[†] | Normal weight (%) | Thinness (%) | | Overweight/Obesity (%) | |
|---|---|---|---|---|---|---|
| | | | n (%) | Crude RRR (95%CI) | n (%) | Crude RRR (95%CI) |
| **Age (years)** | | | | | | |
| 9 | 349 (30) | 269 (77) | 26 (7) | 1.00 | 54 (16) | 1.00 |
| 10 | 397 (34) | 297 (75) | 41 (10) | 1.43 (0.85,2.42) | 59 (15) | 0.99 (0.66,1.50) |
| 11 | 424 (36) | 307 (72) | 54 (13) | 1.68 (1.01,2.78) * | 63 (15) | 0.94 (0.62,1.42) |
| **Sex** | | | | | | |
| Male | 534 (46) | 400 (75) | 62 (12) | 1.00 | 72 (14) | 1.00 |
| Female | 636 (54) | 473 (74) | 59 (9) | 0.78 (0.53,1.15) | 104 (16) | 1.19 (0.85,1.66) |
| **Time spent watching TV/play video games** | | | | | | |
| < = 2 hours | 756 (64) | 570 (65) | 81 (67) | 1.00 | 105 (60) | 1.00 |
| 2 hours+ | 414 (35) | 303 (35) | 40 (33) | 0.91 (0.61,1.37) | 71 (40) | 1.25 (0.89,1.75) |
| **Time spent walking to school** | | | | | | |
| < = 30 minutes | 1023 (87) | 765 (75) | 99 (10) | 1.00 | 159 (16) | 1.00 |
| 30+ minutes | 147 (13) | 108 (74) | 22 (15) | 1.86 (1.10,3.13) * | 17 (12) | 0.89 (0.51,1.56) |
| **Time spent reading** | | | | | | |
| < = 2 hours | 881 (75) | 661 (76) | 92 (76) | 1.00 | 128 (73) | 1.00 |
| 2 hours+ | 289 (25) | 212 (24) | 29 (24) | 1.01 (0.64,1.58) | 48 (27) | 1.20 (0.82,1.74) |
| **Activities before and after school** | | | | | | |
| No | 179 (15) | 123 (69) | 19 (11) | 1.00 | 37 (21) | 1.00 |
| Yes | 991 (85) | 750 (76) | 102 (10) | 0.95 (0.55,1.62) | 139 (14) | 0.66 (0.43,1.02) |
| **Breaktime exercises at school** | | | | | | |
| No | 224 (19) | 157 (70) | 17 (8) | 1.00 | 50 (22) | 1.00 |
| Yes | 945 (81) | 715 (76) | 104 (11) | 1.47 (0.85,2.56) | 126 (13) | 0.61 (0.41,0.89) * |
| **Meat and alternatives** | | | | | | |
| Low | 413 (35) | 305 (35) | 38 (31) | 1.00 | 70 (40) | 1.00 |
| Medium | 377 (32) | 287 (33) | 38 (31) | 1.02 (0.63,1.65) | 52 (30) | 0.76 (0.51,1.13) |
| High | 380 (33) | 281 (32) | 45 (37) | 1.25 (0.78,1.99) | 54 (31) | 0.81 (0.54,1.21) |
| **Fast foods snacks[§]** | | | | | | |
| Low | 510 (44) | 375 (43) | 45 (37) | 1.00 | 90 (51) | 1.00 |
| Medium | 324 (28) | 244 (28) | 31 (26) | 1.03 (0.63,1.69) | 49 (28) | 0.82 (0.55,1.21) |
| High | 336 (29) | 254 (29) | 45 (37) | 1.52 (0.97,2.39) | 37 (21) | 0.63 (0.41,0.96) * |
| **Sugary/sweetened drinks** | | | | | | |
| Low | 408 (35) | 295 (72) | 41 (10) | 1.00 | 72 (18) | 1.00 |
| Medium | 388 (33) | 303 (79) | 34 (9) | 0.78 (0.48,1.26) | 51 (13) | 0.66 (0.44,0.99) * |
| Hiigh | 374 (32) | 275 (74) | 46 (12) | 1.18 (0.75,1.88) | 53 (14) | 0.78 (0.52,1.16) |
| **Fruits and vegetables** | | | | | | |
| Low | 485 (42) | 367 (42) | 46 (38) | 1.00 | 72 (41) | 1.00 |
| Medium | 468 (40) | 353 (40) | 52 (43) | 1.16 (0.76,1.78) | 63 (36) | 0.90 (0.62,1.31) |
| High | 217 (19) | 153 (18) | 23 (19) | 1.16 (0.67,2.00) | 41 (23) | 1.32 (0.85,2.05) |
| **Sweet snacks[a]** | | | | | | |
| Low | 402 (34) | 291 (33) | 39 (32) | 1.00 | 72 (41) | 1.00 |
| Medium | 406 (35) | 306 (35) | 46 (38) | 1.10 (0.69,1.74) | 54 (31) | 0.70 (0.47,1.04) |
| High | 362 (31) | 276 (32) | 36 (30) | 0.95 (0.58,1.55) | 50 (28) | 0.72 (0.48,1.08) |
| **Milk and milk products** | | | | | | |
| Low | 472 (40) | 354 (41) | 48 (40) | 1.00 | 70 (40) | 1.00 |
| Medium | 358 (31) | 270 (31) | 34 (28) | 0.93 (0.58,1.48) | 54 (31) | 1.01 (0.68,1.50) |

(*Continued*)

**Table 1.** (Continued)

| Characteristics | N[†] | Normal weight (%) | Thinness (%) | | Overweight/Obesity (%) | |
|---|---|---|---|---|---|---|
| | | | n (%) | Crude RRR (95%CI) | n (%) | Crude RRR (95%CI) |
| High | 340 (29) | 249 (29) | 39 (32) | 1.13 (0.71,1.80) | 52 (30) | 1.04 (0.69,1.55) |

Significance levels for home level correlates

* p<0.05,

** p<0.01,

*** p<0.001.

RRR relative risk ratio.

CI confidence interval.

[†] N column percentages.

[§] Fast food snacks including any foods such as fried potatoes, bananas, samosas etc.

[a] Sweets snacks including any sugary snacks such as candies, biscuits and cakes.

consumption of sugary/sweetened drinks were at a lower risk of overweight/obesity compared to those who reported low or high levels of consumption (RRR 0.66, 95% CI 0.44,0.99). There were no significant associations between BMIZ categories and the remaining dietary factors.

## Home level correlates

The availability of sugary/sweetened drinks at home was associated with higher risk of overweight or obesity (RRR 1.78, 95%CI 1.22, 2.61). Children who reported availability of television (RRR 2.84, 95%CI 1.65, 4.87) and electronic gadgets such as video games or tablets (RRR1.63, 95%CI 1.11, 2.38) were at higher risk of overweight and obesity compared to children who reported not having these facilities at home (Table 2).

## Home neighborhood level correlates

Table 3 illustrates that (96; 13%) of children who lived in neighborhoods with available playgrounds were at a lower risk of overweight/obesity (RRR 0.63, 95% CI 0.45, 0.88) versus (80; 19%) of children who lived in neighborhoods with no playgrounds.

## School level correlates

Table 4 illustrates school level correlates for overweight and obesity. Among children enrolled 569 (49%) were from urban areas. Children attending schools in urban areas (RRR 3.56; 95% CI 2.22, 5.72) or private schools (4.09; 95% CI 2.53, 6.63) were at higher risk of being overweight or obese compared to those in rural and government schools. Availability of playgrounds in schools (RRR 0.22; 95% CI 0.09, 0.56) was associated with a lower risk of overweight and obesity.

**Adjusted correlates for overweight and obesity.** Table 5 illustrates the multivariable adjusted regression model for nutritional status and child, school, home and neighborhood correlates. After adjustment, urban school location (aRRR 3.76; 95% CI 2.49, 5.68), private school type, (aRRR 4.08; 95% CI 2.66, 6.25), and availability of sugary/sweetened drinks at home (aRRR 1.52; 95% CI 1.01, 2.28) remained significantly associated with higher risks of overweight and obesity. Likewise, availability of playgrounds in schools (aRRR 0.68; 95% CI 0.47, 0.97) was associated with lower risk of overweight and obesity. A medium level consumption of sweet snacks was found to be borderline protective against overweight/obesity (aRRR 0.65 (0.42,1.00). Also, children in urban schools (aRRR 0.56; 95% CI 0.36,0.87) and in private

**Table 2. Home environment correlates for overweight and obesity (N = 1170).**

| Characteristics | N[†] | Normal weight (%) | Thinness (%) | | Overweight/Obesity (%) | |
|---|---|---|---|---|---|---|
| | | | n (%) | Crude RRR (95%CI) | n (%) | Crude RRR (95%CI) |
| **Fast foods at home** | | | | | | |
| No | 157 (13) | 114 (73) | 18 (12) | 1.00 | 25 (16) | 1.00 |
| Yes | 1013 (87) | 759 (75) | 103 (10) | 0.74 (0.43,1.29) | 151 (15) | 0.78 (0.48,1.27) |
| **Sugary/sweetened drinks at home** | | | | | | |
| No | 410 (35) | 327 (80) | 41 (10) | 1.00 | 42 (10) | 1.00 |
| Yes | 760 (65) | 546 (72) | 80 (11) | 1.09 (0.72,1.64) | 134 (18) | 1.78 (1.22,2.61) ** |
| **Milk and milk products at home** | | | | | | |
| No | 376 (32) | 292 (78) | 33 (9) | 1.00 | 51 (14) | 1.00 |
| Yes | 794 (68) | 581 (73) | 88 (11) | 1.25 (0.81,1.92) | 125 (16) | 1.15 (0.80,1.65) |
| **Vegetables and fruits at home** | | | | | | |
| No | 89 (8) | 74 (83) | 6 (7) | 1.00 | 9 (10) | 1.00 |
| Yes | 1081 (92) | 799 (74) | 115 (11) | 1.71 (0.72,4.04) | 167 (15) | 1.65 (0.80,3.39) |
| **Television at home** | | | | | | |
| No | 261 (22) | 213 (82) | 31 (12) | 1.00 | 17 (7) | 1.00 |
| Yes | 909 (78) | 660 (73) | 90 (10) | 0.88 (0.56,1.39) | 159 (18) | 2.84 (1.65,4.87) *** |
| **Electronic gadget at home**[a] | | | | | | |
| No | 920 (79) | 699 (76) | 103 (11) | 1.00 | 118 (13) | 1.00 |
| Yes | 250 (21) | 174 (70) | 18 (7) | 0.58 (0.34,1.00) * | 58 (23) | 1.63 (1.11,2.38) * |
| **Mobile phones**[§] | | | | | | |
| No | 137 (12) | 100 (73) | 10 (7) | 1.00 | 27 (20) | 1.00 |
| Yes | 1033 (88) | 773 (75) | 111 (11) | 1.56 (0.78,3.10) | 149 (14) | 0.77 (0.48,1.24) |
| **Parent encourage exercise** | | | | | | |
| No | 410 (35) | 311 (76) | 34 (8) | 1.00 | 65 (16) | 1.00 |
| Yes | 757 (65) | 559 (74) | 87 (12) | 1.51 (0.99,2.32) | 111 (15) | 1.01 (0.71,1.43) |

Significance levels for home level correlates

* $p < 0.05$,

** $p < 0.01$,

*** $p < 0.001$.

RRR relative risk ratio.

CI confidence interval.

[†] N column percentages.

[a] electronic gadgets including video games and tablets.

[§] Mobile phones owned by any family member.

schools (aRRR 0.49; 95% CI 0.31, 0.77) had a lower risk of thinness. The intra class correlation (ICC) was 0.02, implying that 2% of the variation in nutritional status was explained by the between school variation (data not shown).

## Discussion

We found that overweight and obesity were more prevalent among primary school children in urban areas and in private schools than in rural areas and government schools in Tanzania. Similarly, the prevalence of thinness was more prevalent in rural areas and in government schools. In addition, availability of playgrounds in schools was protective against overweight and obesity while availability of sugary/sweetened drinks at home was an important correlate

**Table 3. Home neighborhood environment correlates for overweight and obesity (N = 1170).**

| Characteristics | N[†] | Normal weight (%) | Thinness (%) | | Overweight/Obesity (%) | |
|---|---|---|---|---|---|---|
| | | | n (%) | Crude RRR (95%CI) | n (%) | Crude RRR (95%CI) |
| **Neighbourhood street vendors** | | | | | | |
| No | 199 (17) | 144 (72) | 23 (12) | 1.00 | 32 (16) | 1.00 |
| Yes | 971 (83) | 729 (75) | 98 (10) | 0.79 (0.48,1.30) | 144 (15) | 0.83 (0.54,1.29) |
| **Neighbourhood playground** | | | | | | |
| No | 414 (35) | 293 (71) | 41 (10) | 1.00 | 80 (19) | 1.00 |
| Yes | 756 (65) | 580 (77) | 80 (11) | 1.02 (0.68,1.53) | 96 (13) | 0.63 (0.45,0.88) ** |
| **Neighbourhood safety** | | | | | | |
| No | 742 (63) | 556 (75) | 75 (10) | 1.00 | 111 (15) | 1.00 |
| Yes | 428 (37) | 317 (74) | 46 (11) | 1.12 (0.75,1.67) | 65 (15) | 1.07 (0.76,1.51) |

Significance levels for neighborhood correlates

* p<0.05,

** p<0.01,

*** p<0.001.

[†] N column percentages.

RRR relative risk ratio.

CI confidence interval.

for childhood overweight and obesity. Medium level consumption of sweet snacks showed a protective effect against overweight/obesity.

Generally, overweight and obesity were more common than thinness although thinness prevalence was still quite high. These findings indicate the existence of the two forms of malnutrition (under nutrition and over nutrition) in Tanzanian primary school children and highlight the importance of considering both forms of malnutrition in future studies. The prevalence of thinness and overweight/obesity observed, were similar to what has been reported recently in a scoping review [26].

Studies done in Nairobi, Arusha and Dar es Salaam reported higher prevalence of overweight and obesity ranging from 18%–25% [12–14], than in our study (15%). Nairobi, Arusha and Dar es Salaam are big cities associated with urbanization and relative sedentary lifestyle. We anticipate that, differences in culture, lifestyle and other environmental aspects e.g., presence of many fast food' outlets, and the use of motorized transport could explain the higher prevalence as compared to what is observed in Moshi (Kilimanjaro), which is a small town, and the nearby rural area. Usually, in Tanzania children from rural and government schools are free to move, play, walk to school and routinely engage in physical activities such as mopping or sweeping the classrooms, gardening etc which helps them maintain a healthy weight.

Children in schools with playgrounds were less likely to be overweight/obese. Reviews and studies on environment and physical activity in high-income countries reported a strong association on availability of playgrounds in schools and high levels of physical activities in children, which is an important modifiable factor for overweight/obesity [27–29].

In the crude analysis, we identified that availability of television and electronic gadgets at home was associated with a higher risk of overweight and obesity. Watching television and playing electronic games are important contributors to a sedentary lifestyle, which makes children reduce their time for physical activity. This is in line with studies conducted in Dar es Salaam, and a multinational study of 12 countries, which reported an association of screen time with overweight/obesity [11,30]. Increase in screen time is attributed to globalization, which promotes adaptations to technological advancements with more uses of television and

**Table 4. School level correlates for overweight and obesity (N = 1170).**

| Characteristics | N[†] | Normal weight (%) | Thinness (%) | | Overweight/Obesity (%) | |
|---|---|---|---|---|---|---|
| | | | n (%) | Crude RRR (95%CI) | n (%) | Crude RRR (95%CI) |
| **School location** | | | | | | |
| Moshi Rural | 601 (51) | 476 (79) | 80 (13) | 1.00 | 45 (8) | 1.00 |
| Moshi Urban | 569 (49) | 397 (70) | 41 (7) | 0.63 (0.38,1.04) | 131 (23) | 3.56 (2.22,5.72) *** |
| **School ownership/type** | | | | | | |
| Government | 589 (50) | 467 (79) | 84 (14) | 1.00 | 38 (7) | 1.00 |
| Private | 581 (50) | 406 (70) | 37 (6) | 0.50 (0.30,0.82) ** | 138 (24) | 4.09 (2.53,6.63) *** |
| **Availability of playgrounds** | | | | | | |
| No | 1121 (96) | 847 (76) | 119 (11) | 1.00 | 155 (14) | 1.00 |
| Yes | 49 (4) | 26 (53) | 2 (4) | 1.77 (0.35,8.91) | 21 (43) | 0.22 (0.09,0.56) ** |
| **Availability of sports equipment** | | | | | | |
| No | 834 (71) | 626 (75) | 90 (11) | 1.00 | 118 (14) | 1.00 |
| Yes | 336 (29) | 247 (74) | 31 (9) | 1.12 (0.62,2.03) | 58 (17) | 0.79 (0.46,1.34) |
| **Consume food from street vendors** | | | | | | |
| No | 216 (18) | 167 (77) | 26 (12) | 1.00 | 23 (11) | 1.00 |
| Yes | 954 (82) | 706 (74) | 95 (10) | 1.15 (0.60,2.23) | 153 (16.0) | 0.64 (0.33,1.23) |
| **Street vendors permitted within school surroundings** | | | | | | |
| No | 345 (29) | 256 (74) | 33 (10) | 1.00 | 56 (16) | 1.00 |
| Yes | 825 (71) | 617 (75) | 88 (11) | 0.95 (0.52,1.74) | 120 (15) | 1.19 (0.68,2.06) |

Significance levels for school level correlates

* p<0.05,

** p<0.01,

*** p<0.001.

[†] N column percentages.

RRR relative risk ratio.

CI confidence interval.

electronic gadgets. Interventions to promote physical activity should aim at reducing screen time for school children and encouraging children to be more involved in regular physical activities.

Despite diet being one of the important correlates for nutritional status (overweight/obesity), only consumption of a sugary/sweetened drinks at home was significantly associated with overweight/obesity while reported medium level consumption of sweet snacks was found to be protective against overweight/obesity. This observation was not expected as generally consumption of sweet snacks is associated with overweight/obesity [31]. Other dietary characteristics showed a weak or no evidence of an association with overweight and obesity. We expected to gain some knowledge on the effects of diet, as consumption of a healthy diet which is less energy dense, includes enough fruits and vegetables, fewer sweet snacks and fast foods, is linked with lower risks of overweight/obesity [32]. Our study is in line with studies done in China and a recent review done by ISCOLE team [33,34] which only documented a weak relationship with any diet pattern scores and overweight/obesity. Conversely, a recent study in Mozambique from ISCOLE team reported a positive association between fast food intake and overweight/obesity [10]. The use of a qualitative food frequency questionnaire rather than a quantitative one with detailed information on true food intake may have contributed to the lack of clear associations between diet and body weight. Also, other well-known challenges with dietary assessment methods such as recall bias, misreporting or under-reporting might

**Table 5. §Adjusted correlates for overweight and obesity (N = 1170).**

| Characteristics | Thinness (%) | Overweight/Obesity (%) |
|---|---|---|
| | aRRR (95%CI) | aRRR (95%CI) |
| **Age (years)** | | |
| 9 | 1.00 | 1.00 |
| 10 | 1.26 (0.74,2.14) | 1.22 (0.78,1.91) |
| 11 | 1.72 (1.03,2.87) * | 1.14 (0.72,1.78) |
| **Sex** | | |
| Male | 1.00 | 1.00 |
| Female | 0.74 (0.50,1.10) | 1.20 (0.84,1.73) |
| **School location** | | |
| Moshi Rural | 1.00 | 1.00 |
| Moshi Urban | 0.56 (0.36,0.87) * | 3.76 (2.49,5.68) *** |
| **School ownership/type** | | |
| Government | 1.00 | 1.00 |
| Private | 0.50 (0.31,0.79) ** | 4.08 (2.66,6.25) *** |
| **Walking to school** | | |
| No | 1.00 | 1.00 |
| Yes | 1.49 (0.87, 2.53) | 0.99 (0.55, 1.79) |
| **Breaktime exercise** | | |
| No | 1.00 | 1.00 |
| Yes | 1.08 (0.61, 1.90) | 0.78 (0.51,1.17) |
| **Fruits and vegetables** | | |
| Low | 1.00 | 1.00 |
| Medium | 0.14 (0.57,1.78) | 0.97 (0.64,1.47) |
| High | 1.32 (0.75, 2.32) | 1.30 (0.81,2.09) |
| **Fast foods at home** | | |
| No | 1.00 | 1.00 |
| Yes | 0.94 (0.53,1.68) | 0.61 (0.36,1.06) |
| **Sugary/sweetened drinks at home** | | |
| No | 1.00 | 1.00 |
| Yes | 1.45 (0.94,2.22) | 1.52 (1.01,2.28) * |
| **Sweet snacks** | | |
| Low | 1.00 | 1.00 |
| Medium | 1.14 (0.71,1.85) | 0.65 (0.42,1.00) |
| High | 1.02 (0.60, 1.72) | 0.64 (0.40,1.00) |
| **Availability of playgrounds** | | |
| No | 1.00 | 1.00 |
| Yes | 0.88 (0.58,1.34) | 0.68 (0.47,0.97) * |

§ Adjusted for age, sex, school location, school type, time spent watching television, availability of playgrounds at school, fast foods at home, sugary/sweetened drinks at home.

Significance levels for child, school and home level correlates

* p<0.05,

** p<0.01,

*** p<0.001.

aRRR adjusted relative risk ratio.

CI confidence interval.

have contributed to these poor associations [35–37]. Furthermore, it is possible that those children who reported eating fast food were consuming only very small amounts that could not contribute to any health outcome unless consumed over a long period of time. In Tanzania fast food is expensive.

The main strengths of this study are its large sample size and the use of a socio-ecological approach to understand multiple factors that contribute to childhood overweight and obesity in urban and rural districts. However, this study has some limitations: the cross-sectional nature of the study meant we could not assess causality; and self-reports of diet and physical activity are prone to recall bias. Also, the exclusion of children who could not read and write may have led to selection bias, as these children might have different rates of overweight/obesity or thinness. Further there is a potential limitation to conducting subgroup analysis due to their smaller sample size.

## Conclusion

The study highlights important correlates of overweight and obesity. Efforts should be undertaken to encourage physical activities in primary school children particularly in private schools and in urban settings. Availability of playgrounds in schools is essential in promoting more engagement in physical activity.

## Supporting information

**S1 Dataset. Childhood obesity.**
(DTA)

**S1 File. Questionnaire English version.**
(DOCX)

**S2 File. Questionnaire Swahili version.**
(DOC)

## Acknowledgments

We thank Heiner Grosskurth, research staff for their support during data collection and field arrangements. We acknowledge the valuable input of children and teachers during their participation in this study.

## Author Contributions

**Conceptualization:** Mary Vincent Mosha, Sia E. Msuya, Suzanne Filteau.

**Data curation:** Mary Vincent Mosha, Sia E. Msuya, Elizabeth Kasagama, Philip Ayieko, Jim Todd, Suzanne Filteau.

**Formal analysis:** Mary Vincent Mosha, Elizabeth Kasagama, Philip Ayieko, Jim Todd.

**Funding acquisition:** Mary Vincent Mosha, Sia E. Msuya, Suzanne Filteau.

**Investigation:** Mary Vincent Mosha.

**Methodology:** Mary Vincent Mosha, Sia E. Msuya, Elizabeth Kasagama, Philip Ayieko, Jim Todd, Suzanne Filteau.

**Project administration:** Mary Vincent Mosha, Sia E. Msuya, Suzanne Filteau.

**Resources:** Mary Vincent Mosha.

**Software:** Mary Vincent Mosha, Elizabeth Kasagama, Philip Ayieko, Jim Todd.

**Supervision:** Sia E. Msuya, Suzanne Filteau.

**Validation:** Mary Vincent Mosha, Sia E. Msuya, Elizabeth Kasagama, Philip Ayieko, Jim Todd, Suzanne Filteau.

**Visualization:** Mary Vincent Mosha, Sia E. Msuya, Elizabeth Kasagama, Philip Ayieko, Jim Todd, Suzanne Filteau.

**Writing – original draft:** Mary Vincent Mosha.

**Writing – review & editing:** Mary Vincent Mosha, Sia E. Msuya, Elizabeth Kasagama, Philip Ayieko, Jim Todd, Suzanne Filteau.

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
