## [Decision Letter · Decision Letter 0]

6 Jan 2021

PONE-D-20-30826

Prevalence and correlates of overweight and obesity among primary school children in Kilimanjaro, Tanzania

PLOS ONE

Dear Dr. Mosha,

Thank you for submitting your manuscript to PLOS ONE. After careful consideration, we feel that it has merit but does not fully meet PLOS ONE’s publication criteria as it currently stands. Therefore, we invite you to submit a revised version of the manuscript that addresses the points raised during the review process.

We look forward to receiving your revised manuscript.

Kind regards,

Sabine Rohrmann

Academic Editor

PLOS ONE

Journal Requirements:

2. Please include additional information regarding the survey or questionnaire used in the study and ensure that you have provided sufficient details that others could replicate the analyses.

For instance, if you developed a questionnaire as part of this study and it is not under a copyright more restrictive than CC-BY, please include a copy, in both the original language and English, as Supporting Information.

3. Please provide additional details regarding participant consent.

In the ethics statement in the Methods and online submission information, please ensure that you have specified whether you obtained consent from parents or guardians. If the need for consent was waived by the ethics committee, please include this information.

4. Please amend the manuscript submission data (via Edit Submission) to include authors Sia E. Msuya, Elizabeth Kasagama, Philip Ayieko, Jim Todd, Heiner Grosskurth and Suzanne Filteau.

Reviewers' comments:

Reviewer's Responses to Questions

**Comments to the Author**

1. Is the manuscript technically sound, and do the data support the conclusions?

Reviewer #1: Yes

2. Has the statistical analysis been performed appropriately and rigorously? 

Reviewer #1: Yes

3. Have the authors made all data underlying the findings in their manuscript fully available?

Reviewer #1: Yes

4. Is the manuscript presented in an intelligible fashion and written in standard English?

Reviewer #1: Yes

5. Review Comments to the Author

Reviewer #1: The authors present an important issue on childhood overweight and obesity in SSA/Tanzania. The data presentation is logical and easy to follow. However, there are some issues that need to be addressed to improve the report for better readership understanding.

1. The short title is similar to the full title, need to be edited/re-written

2. The authors studied 9-11 year olds, however there is no place they explained why limit was put to this population only? they need to explain the basis of choosing this sample population, and not for example 6-15/16 which is the primary school-age group in Tanzania

3. Could the authors provide the total number of children first in the area/region studied, then also the number of all pupils in the schools studied, and provide the proportion studied. This will give the reader a better picture of the representativeness.

4. Consent was given by school administrators and assent by children. How was assent provided? written or verbal? it has to be stated.

5. Since the WHO categorization included obesity and overweight as separate entities, could the authors provide how many children were obese? what was the prevalence of obesity and overweight as separate?

6. Looking at table 1, there are variables that were associated with overweight and obesity BUT without scientific explanations, eg. children who reported medium consumption of sugary/sweetened drinks were at a lower risk of overweight/obesity (RRR 0.66), high fast food food/snacks associate with lower obesity/overweight. Authors should at least explain. This brings my next question/comment

7. Was the study powered to look at the significance of the different correlates? as the sample size was powered to determine the prevalence only. Authors should explain their assumptions for the variables they analyzed. Otherwise, this should be a limitation. I feel there were few outcomes for some/most of the variables and the study could have been less powered to see these associations.

8. Because thinness was not factored in the sample size calculation, I feel it also came as an additional burden to the sample, in terms of the power to see the associations. Authors should include this in their limitation.

9. About the tool used to assess the food intake/behaviours at home. I wonder whether this tool was culturally adapted before being used in the current study? Authors should mention, whether there was formal adaptation duirng this study or from previous studies/researchers. This is important, and maybe explain some of the variations seen in responses. What is relevant in developed/Europe/America countries may not be relevant in Tanzania. Please explain how was the tool translated into Swahili and if adaptation was done or not.

10. One of the most important driver of childhood obesity is parents' socio-economic status, which mainly explain the factors also found in this study. Please explain in the methods why this was not put in the context.

6. PLOS authors have the option to publish the peer review history of their article (what does this mean?). If published, this will include your full peer review and any attached files.

Reviewer #1: No

---

## [Author Response · Author response to Decision Letter 0]

18 Feb 2021

PONE-D-20-30826

Prevalence and correlates of overweight and obesity among primary school children in Kilimanjaro, Tanzania

PLOS ONE

Addressed journal requirements and reviewer’s comments 

 Responses to journal requirements

1. Please ensure that your manuscript meets PLOS ONE's style requirements, including those for file naming. The PLOS ONE style templates can be found 

athttps://journals.plos.org/plosone/s/file?id=wjVg/PLOSOne_formatting_sample_main_body.pdf and

Response: We revised the manuscript to meet the journal style requirements. 

2. Please include additional information regarding the survey or questionnaire used in the study and ensure that you have provided sufficient details that others could replicate the analyses.

For instance, if you developed a questionnaire as part of this study and it is not under a copyright more restrictive than CC-BY, please include a copy, in both the original language and English, as Supporting Information.

Response: We included additional information regarding the questionnaire used. Both sets of questionnaires are attached as supporting information (The English version and a Swahili version)

3. Please provide additional details regarding participant consent.

In the ethics statement in the Methods and online submission information, please ensure that you have specified whether you obtained consent from parents or guardians. If the need for consent was waived by the ethics committee, please include this information. 

Response: We updated the ethics section: we describe that consent was obtained from head teachers / school administrators and that children gave the written assent. We experienced difficulties in obtaining written parental consent as some parents did not return the consent forms that we had sent to them through their children, but instead only gave verbal consent. To address this problem, we consulted the ethics committee which provided a waiver: because the study did not involve invasive procedures, the committee allowed headteachers / administrators to consent for school children. In addition, before starting field work, children were sent home with information sheets on the study to hand out to their parents. Furthermore, before enrolment in the study and after the study had been explained to them, children were asked to sign a written assent form if they wished to take part. (please refer to lines 232 – 241)

4. Please amend the manuscript submission data (via Edit Submission) to include authors Sia E. Msuya, Elizabeth Kasagama, Philip Ayieko, Jim Todd, Suzanne Filteau.

 Response: These authors are now included in the manuscript submission system

Response: The caption for each figure is now included (please refer to line 142).

Responses to Questions raised by the reviewer 

Reviewer #1: The authors present an important issue on childhood overweight and obesity in SSA/Tanzania. The data presentation is logical and easy to follow. However, there are some issues that need to be addressed to improve the report for better readership understanding.

1. The short title is similar to the full title, need to be edited/re-written

Response: The short title is now included in the manuscript submission system “Overweight and obesity among school children in Tanzania”

2. The authors studied 9-11 year old’s, however there is no place they explained why limit was put to this population only? they need to explain the basis of choosing this sample population, and not for example 6-15/16 which is the primary school-age group in Tanzania

Response: We have incorporated the rationale for this approach in the Methods section. We selected age group 9 -11 because by the age of 9 children were old enough to read and write and express themselves. Regarding the upper end of the age range, the study focused on a tight age range as there is a lot of change over the ages 6 to 15 years. The age range 9-11 represents a group of children who are more homogenous in their physical activity (please refer to lines 106 – 113).

3. Could the authors provide the total number of children first in the area/region studied, then also the number of all pupils in the schools studied, and provide the proportion studied. This will give the reader a better picture of the representativeness.

Response: The requested information has now been provided (please refer to line 100 -102).

4. Consent was given by school administrators and assent by children. How was assent provided? written or verbal? it has to be stated.

Response: As explained under point 3 above, we obtained written assent from children after explaining the study in detail to them. Children had the right to participate or not to participate in the study (please refer to lines 232 – 241).

5. Since the WHO categorization included obesity and overweight as separate entities, could the authors provide how many children were obese? what was the prevalence of obesity and overweight as separate?

Response: We have now included information on the prevalence of overweight and obesity separately (please refer lines 30, 246).

6. Looking at table 1, there are variables that were associated with overweight and obesity BUT without scientific explanations, eg. children who reported medium consumption of sugary/sweetened drinks were at a lower risk of overweight/obesity (RRR 0.66), high fast-food food/snacks associate with lower obesity/overweight. Authors should at least explain. This brings my next question/comment

Response: Thanks for this observation. In Tanzania, the consumption of items such as chicken and chips, sodas or other sugary drinks is considered as a sign of wealth. Therefore, responses may have been affected by social desirability bias. We agree the association is as unexpected. However, in the adjusted analysis it only borderline significant and may just have occurred due to chance, as can happen particularly when many variables are examined. 

7. Was the study powered to look at the significance of the different correlates? as the sample size was powered to determine the prevalence only. Authors should explain their assumptions for the variables they analyzed. Otherwise, this should be a limitation. I feel there were few outcomes for some/most of the variables and the study could have been less powered to see these associations.

Response: During planning this study, there was a limited number of papers available that looked at overweight and obesity in African children. That is why our sample size calculation was driven by assumptions on the prevalence of overweight and obesity. We acknowledge that the study might have been under - powered to detect association between diet, physical activity and our key outcome (overweight and obesity) and thus type 2 errors might have been introduced.

We conducted exploratory analysis on the factors associated with overweight and obesity in this age group in Tanzania. Most studies have focused on socio demographics as risk factors, but this study included data on diet and physical activity, which are on the causal pathway to overweight in children in other countries. Although the study is not powered for the exploratory analysis, the results do give pointers for future studies which will be conducted in this population. 

8. Because thinness was not factored in the sample size calculation, I feel it also came as an additional burden to the sample, in terms of the power to see the associations. Authors should include this in their limitation.

Response: Thank you for the comment. We actually think that also looking at thinness is not a limitation but a strength of our study. Thinness did not affect the sample size required to detect overweight / obesity with the desired precision. – Independent of this, we included information on the potential limitation of subgroup analyses such as in this case, as the number of children who were thin was small.

9. About the tool used to assess the food intake/behaviors at home. I wonder whether this tool was culturally adapted before being used in the current study. Authors should mention whether there was formal adaptation during this study or from previous studies/researchers. This is important, and maybe explain some of the variations seen in responses. What is relevant in developed/Europe/America countries may not be relevant in Tanzania. Please explain how the tool was translated into Swahili and if adaptation was done or not.

Response: For this study, we adapted and modified the tool which had been used in the multicenter study on childhood overweight / obesity, environment and lifestyle which involved Kenya as one of the low - and middle - income countries. This information is now elaborated in the manuscript (please see lines 145 – 153).

10. One of the most important driver of childhood obesity is parents' socio-economic status, which mainly explain the factors also found in this study. Please explain in the methods why this was not put in the context.

Response: We acknowledge that many studies especially in high income countries have found an association between parental social economic status (SES) and overweight / obesity. However, this study did not collect this information because during the pilot study we observed that children were not able to answer questions to assess SES. Furthermore, we experienced difficulties in reaching parents during the pilot study which was a major limitation and made us to not include them in the main study. However, attendance at private schools is highly associated with higher SES and therefore we used enrollment at private or government schools as a proxy indicator for SES. (please refer to lines 129 – 130).

---

## [Editor Report · Decision Letter 1]

22 Mar 2021

Prevalence and correlates of overweight and obesity among primary school children in Kilimanjaro, Tanzania

PONE-D-20-30826R1

Dear Dr. Mosha,

We’re pleased to inform you that your manuscript has been judged scientifically suitable for publication and will be formally accepted for publication once it meets all outstanding technical requirements.

Kind regards,

Sabine Rohrmann

Academic Editor

PLOS ONE
---

## [Editor Report · Acceptance letter]

14 Apr 2021

PONE-D-20-30826R1 

Prevalence and correlates of overweight and obesity among primary school children in Kilimanjaro, Tanzania 

Dear Dr. Mosha:

I'm pleased to inform you that your manuscript has been deemed suitable for publication in PLOS ONE. Congratulations! Your manuscript is now with our production department. 

Kind regards, 

on behalf of

Dr. Sabine Rohrmann 

Academic Editor

PLOS ONE